# Unprecedent cave ice melt in the last 6100 years in

# the Central Pyrenees (A294 ice cave)

- Carlos Sancho<sup>1†</sup>, Ánchel Belmonte<sup>2</sup>, Maria Leunda<sup>3,\_4,5</sup>, Marc Luetscher<sup>46</sup>,
- Christoph Spötl<sup>57</sup>, Juan Ignacio López-Moreno<sup>68</sup>, Belén Oliva-Urcia<sup>1</sup>, Jerónimo
- López-Martínez<sup>79</sup>, Ana Moreno<sup>68</sup>, Miguel Bartolomé\*\*\*\*6, 10, 11

- 1 Departamento de Ciencias de la Tierra, Universidad de Zaragoza, C/ Pedro Cerbuna,
- 12, 50009 Zaragoza, Spain
- 2 Sobrarbe-Pirineos UNESCO Global Geopark, Avda. Ordesa 79, 22340 Boltaña, Spain
- 3 Department of Plant Biology and Ecology, Faculty of Science and Technology,
- University of the Basque Country (UPV EHU), Barrio Sarriena s/n, 48940, Leioa, Bizkaia,
- Spain
- 4 Swiss Federal Institute for Forest, Snow and Landscape Research WSL
- Zürcherstrasse 111, 8903 Birmensdorf, Switzerland
- <u>35</u> Institute of Plant Sciences & Oeschger Centre for Climate Change Research.
- University of Bern, Alternbergrain 21, 3012, Bern, Switzerland
- 46 Swiss Institute for Speleology and Karst Studies (SISKA), Rue de la Serre 68, 2300
- La Chaux-de-Fonds, Switzerland
- <u>57</u> Institute of Geology, University of Innsbruck, Innrain 52, 6020 Innsbruck, Austria
- 68 Departamento de Procesos Geoambientales y Cambio Global, Instituto Pirenaico de
- Ecología (CSIC), Avda. Montañana 1005, 50059 Zaragoza, Spain
- <u>79</u> Facultad de Ciencias, Universidad Autónoma de Madrid, Francisco Tomás y Valiente
- 7, 28049 Madrid, Spain
- 10 Geological Institute, NO G59, Department of Earth Sciences, Sonneggstrasse 5,
- ETH, 8092 Zurich, Switzerland
- 811 Departamento de Geología, Museo Nacional de Ciencias Naturales (CSIC), C/ de
- José Gutiérrez Abascal, 2, 28006 Madrid, Spain
- † Deceased
- \*Correspondence to: mbart@mncn.csic.es

# ABSTRACT

40

43

45

50

59 Ice caves are understudied environments within the cryosphere, hosting unique ice deposits valuable for paleoclimate studies. Recently, many of these deposits have experienced accelerated retreat due to global warming, threatening their existence. The A294 cave contains the world's known oldest firn cave deposit (6100 years yr cal- BP), which is progressively waning. This study presents 12 years (2009-2021) of monitoring data from of A294 cave, including temperature measurements both outside and inside the cave, meteoric precipitation, and ice loss measurements by comparing historical cave surveys (1978, 2012, 2019), photographs, and ice measurements within the cave. Our findings indicate a continuous increase in cave air temperature (~1.07 to 1.56 °C over 12 years), increases in the Thaw Index, and a decrease in the number of freezing days (i.e., days below 0 °C) as well as in the Freezing Index. Calculated melting rates based on cave surveys and measurements show significant variations depending on the cave sector, ranging from ~15 to ~192 cm per year. The retreat of the ice body is primarily driven by an increase in winter temperatures, the rise in more rainfall during the warm seasons, and the a decrease in snowfall and snow cover duration. The ice stratigraphy and local paleoclimate records suggest unprecedented melting conditions since this ice began to form about 6100 years ago. This study highlights the urgent need to recover all possible information from these unique subterranean ice deposits before they disappear.

# 1. Introduction

The mean global temperature has risen by 0.8 to 1.1 °C since the early 20<sup>th</sup> century (IPCC, 2021). However, specific mountainous regions in Southern Europe, such as the Pyrenees, have experienced a temperature surge of ~1.3 °C since 1949 (Observatorio Pirenaico de Cambio Global, 2018), indicating an a recent acceleration in warming nearly twice as much as the global average. This temperature increase is directly impacting the glaciers in the Pyrenees, which have receded significantly since the end of the Little Ice Age (González Trueba et al., 2008; García-Ruiz et al.,

2014). The pace of this retreat has notably accelerated in recent years (López-Moreno et al., 2016; Vidaller et al., 2021, 2023). Glaciers are not the only elements of the cryosphere affected by global warming: Mountain permafrost and ice caves are also facing critical threats (Kern and Perşoiu, 2013; Biskaborn et al., 2019), with ice caves representing the least studied systems.

The scientific community has recognized the pressing need for a comprehensive research initiative aimed at recovering untapped physical, chemical, and biological information stored within cave ice bodies prior to their complete disappearance (Kern and Perşoiu, 2013). In recent years, the growing scientific interest on ice caves—driven by their rapid degradation—has led to a notable increase in interdisciplinary research efforts (e.g., Barabach and Stasiewicz, 2025n.d.) Unlike glaciers, mid-latitude ice caves are less vulnerable to summer temperature fluctuations and thus have the potential to preserve paleoclimate proxy data information over longer periods. Ice caves have therefore attracted a great deal of attention for research into this unique repository of past climatic conditions (Luetscher et al., 2005; Stoffel et al., 2009; Feurdean et al., 2011; Žák et al., 2012; Perşoiu et al., 2017; Sancho et al., 2018; Leunda et al., 2019; Racine et al., 2022a; Racine et al., 2022b; Bartolomé et al., 2023; Ruiz-Blas et al., 2023).

In static ice caves (i.e. sag type), the ingress of cold and dense air during the winter season cools the ice cave (open phase), while during the warm season, the cave operates as a cold trap (closed phase), with the influence of higher temperatures being nearly negligible (Luetscher et al., 2008). The ice retreat within these ice caves is predominantly regulated by diminished winter precipitation and rising winter temperatures (Luetscher et al., 2005, 2008). The heat exchanged during winter determines the cave temperatures during the closed phase thereby affecting ice melting (Wind et al., 2022). Additionally, the increase in rainfall and extreme precipitation events during warmer seasons (closed phase in ice caves) leads to ice loss due to the release of heat through the penetration of water into the cave environment (Luetscher et al., 2008; Perşoiu et al., 2021). In static ice caves (i.e., sagtype), ice caves, cold and dense air enters the cave during the winter season (open

phase), cooling the cave environment. During the warm season (closed phase), the cave acts as a thermal trap, largely isolating the interior from external temperature fluctuations (Luetscher et al., 2008). However, although external temperatures have minimal direct influence during the closed phase, the thermal state of the cave is still affected by the heat exchange that occurred during the preceding open phase (Wind et al., 2022). This residual heat influences the cave's internal temperature and, consequently, the rate of ice melting. Additionally, increased rainfall and extreme precipitation events during the closed phase contribute to ice loss, as infiltrating water releases heat into the cave environment (Luetscher et al., 2008; Perşoiu et al., 2021). Overall, ice retreat in these caves is primarily driven by reduced winter precipitation, rising winter temperatures, and the thermal effects of water infiltration during warmer months (Luetscher et al., 2005, 2008).

Monitoring the current cave dynamics helps to understand how caves work and thus to recognize the factors that influence the melting of cave ice (Luetscher et al., 2008; Perşoiu et al., 2021; Wind et al., 2022; Bartolomé et al., 2023).

In this study, we present a comprehensive dataset derived from a 12-year monitoring campaign (2009-2021) in A294 ice cave, situated in the Central Pyrenees. The rise in cave temperature and the increase in summer and fall precipitation pose a significant threat to the oldest documented firn in the world, which is preserved in a cave and is between 6100 and 1880 years old (Sancho et al., 2018). The data we have gathered and the analysis of ice stratigraphy and climate reconstructions since the onset of the Neoglacial period provide compelling evidence that the ongoing melting of this exceptional ice body represents the most significant retreat since its accumulation began some 6100 years ago.

### 2. The A294 ice cave

This ice cave is located in the Central Pyrenees (Belmonte-Ribas et al., 2014), in the north-eastern part of the Iberian Peninsula (Fig. 1a). The cave is situated in the Cotiella massif, at an elevation of 2238 m a.s.l. This massif belongs to the Central South Pyrenean Unit and it consists mainly of Upper Cretaceous limestones as well as Eocene limestones and marls (Supplementary material, Fig. S1). The highest point

in this area is the Cotiella peak at 2912 m a.s.l. (Fig. 1b). The Armeña cirque is characterized by periglacial and karst landforms (Fig. 1b). In this region, there is a climatic transition between the Atlantic and Mediterranean mountain climates. The mean annual air temperature (MAAT) at the nearby Armeña Weather weather Station-station (AWS, Figs. 1b, c, d) at 2200 m a.s.l. elevation averages ~5.6 °C (2009-2021; Fig. 1c). The wettest periods in this area are in April/May and October/November. Winters and summers are typified by cold and warm/dry periods, respectively (Fig. 1c). During the colder months, there are more than 20 snowfalls a year between November and April. In the Central Pyrenees, at 2000 m a.s.l. the average snow depth typically ranges between 20 and 80 cm (1985-1999) with annual peaks that often exceed 150 cm. Superimposed on a very large interannual variability, the duration of the snow cover and the accumulation of snow depth have decreased statistically significant since the middle of the 20<sup>th</sup> century (López-Moreno, 2005; López-Moreno et al., 2020). At higher altitudes, around 2600 m a.s.l., the snow depth at the end of winter is often more than 3 m (López-Moreno et al., 2019).

123

130

Figure 1. a) Location of the Cotiella massif in the Central Pyrenees and the different paleoclimate records and the weather station used in this study (1: A294 ice cave (A294), 2: Armeña Weather weather Station (AWS), 3: Basa de la Mora (lake, BSM), 4: Monte Perdido glacier (MP glacier), 5: Redó lake). b) Geomorphological map and location of the A294 ice cave and the weather station (modified from

Belmonte, 2014). c) Monthly mean temperature (red line), maximum and minimum mean temperatures (grey shaded area), annual mean temperature (dashed red line) and mean monthly rainfall (blue bars) over the last 12 years recorded at the Armeña Weather weather Station (AWS) located at 2200 m a.s.l., ~400 m east of the A294 cave entrance (d).

The A294 cave is a sag-type cave, with a depth of ~40 m and two entrances (Fig. 2a, b). The primary entrance consists of a vertical shaft that is ~10 m high-deep and spans an area of roughly 30 m². There is also a second, smaller shaft that is typically closed off by snow in the winter months. The primary entrance is connected via a snow ramp to the cave main chamber, which contains an old ice body (Fig. 2a). The thickness and extent of the snow ramp varies from year to year, depending on the frequency and intensity of snowfall (Belmonte-Ribas et al., 2014). Open conditions marked by a "chimney effect" characterize the winter phase (November-May). Ventilation occurs primarily via the main shaft, while the smaller shaft is usually blocked by snow during fall, winter and sometimes in spring. Conversely, the cave acts as a cold air trap during the closed phase (June-October) (Fig. 2b). Two areas in the cave host seasonal ice stalagmites (IS 1 and 2) underneath several active dripping points (Fig. 2a).

Figure 2: a) Geomorphological map superimposed on the 2019 cave survey. The locations of the temperature sensors is are indicated in by small red rectangles.

The light blue colour represents seasonal snow and the dark blue colour represents the fossil ice body. IS1 and IS2 indicate seasonal ice stalagmites. b) Cave profile with a simplified model of air circulation during winter (open phase, left) and summer (closed phase, right). Blue arrows show the movement of cold air masses, while orange arrows represent warm air fluxes inside the cave. Photo: David Serrano.

157

160

161162

167

171

173

#### 3. Material and methods

#### 3.1 Ice extent

The ice retreat was assessed on the basis of cave surveys from different years (1978, 2012, 2019, Figs. 3a, b, c, d). Following Luetscher et al. (2005) we use a comparison of historical surveys to highlight changes over time, as it is a simple method to observe ice changes. The surveys of 1978 (Grupo de Espeleología Cataluña Aragón, G.E.C.A.) and 2012 (Belmonte-Ribas et al., 2014) were carried out with a normal compass and clinometer. The North arrow in the original 1978 survey was not correctly positioned, and the crevasse between ice and bedrock is questionable, as in 2008 the old ice on the southern wall was in contact with the bedrock (Fig. S2). The crevasse must have been small, because the new gallery (called Gallery 90s) was found during the 1990s when after the ice had retreated far enough (Fig. 2). -The survey in 2019 (Fig. 2) was conducted using a DistoX2-Leica DistoX2 (X310) modified with an electronic base plate to measure the direction and dip (Heeb, 2014). The the precision associated with each measurement is ±1.5 mm. A total of 68 points were measured emphasizing the cave morphology and the ice perimeter. To determine the horizontal retreat of the ice body, the ice perimeter was determined at three reference points/sectors defined and well identified in the field in (Fig. 3 (number 1 to 3)). Potential errors may arise from the inclination of the laser line when taking the measurement. Normally, sSeveral measurements arewere taken to ensure that there are no major variations. This human error could result in inaccuracies of up to 5 cm, which, for distances greater than 10 m, would represent less than 0.5% error (measurements from the Northern wall (1)

and the Ice Ffront (3), (Fig. 3d), and slightly more for the Rock €C∈orner (2). Errors in topographic comparisons depend on the level of detail, the surveyor's expertise, and inherent human measurement inaccuracies. However, the extent of ice loss is so pronounced that even when using only a rough outline of the cave, the measurements obtained from successive surveys are consistent with those reported by Belmonte et al. - (2014). Complementary field measurements using fixed reference points (blocks, walls, dripping points) together with several photographs taken between 2008 and 2023 were also used to reconstruct the changes in size of the ice body. For the Northern sectorwall (1), the three cave surveys were compared. In contrast, for the Rock Ccorner (2) and the Ice Ffront (3), the data consisted of a combination of cave surveys comparisons and field measurements. -To evaluate vertical changes of the ice body during the monitored period, two stratigraphic logs were made (Sancho et al., 2018; Leunda et al., 2019) (Fig. S3). In 2011, when the deposit was first sampled, the ice was 9.25 m thick and decreased to 7.90 m in 2015 when the second stratigraphic log was prepared. The internal structure of the ice sequence also changed between 2011 and 2015 (Leunda et al., 2019) as shown by the cross-stratified ice beds formed during snow deposition. Nevertheless, the identification of a same detrital layer (observed at 165 cm depth from the top in 2015 and at 240 cm depth in 2011) allows comparing correlating the depth-age models for both years (Fig. S3). The spatio-temporal reproducibility of the depth-age models together with the good correspondence between two independently dated terrestrial plant macrofossil remains reveals that, although decadal-scale melting periods may have occurred repeatedly, the macrofossil remains preserved in the detrital layers did not suffer significant movements within the ice deposit. Complementary field measurements using fixed reference points (blocks, walls, dripping points) together with several photographs taken ice body.

179

181

187

191

194

198

Figure 3: Three plan view maps of the A294 ice cave from based on different surveys. a) Corrected cave topography from 1978 ((G.E.C.A), see Fig. S2 for details). \*Note: The dashed galleries on the 1978 survey indicate a cave passage discovered in 2012 and the corrected position of the southern gallery, respectively. b) Cave survey from 2012 (Belmonte et al., 2014). c) Cave survey from 2019 (this study). The insets, marked with black dashed circles, correspond to sectors where topographic changes were measured and indicated by numbers: (1) Northern wall, (2) Rock Ceorner (green point) and (3) Ice front. Red arrows correspond to distances selected for measurements on the cave surveys, and some of them in the field. d) Photograph taken from the main shaft, showing the approximate position of the ice body between 1978 and 2019.

### 3.2 Environmental monitoring

211212

219

225226

231232

The cave air temperature was recorded over 12 years (2009-2021) (Fig. 4) using Hobo Pro v2 U23-001 sensors with an accuracy of ±0.21 °C and a resolution of 0.02 °C at 0 °C. In 2009, six sensors were installed in the cave, of which the three time series with the longest and most complete temperature records (CH1, CH2 and CH3) were used for this study (same labels as in Belmonte-Ribas et al., 2014). The sensors were set up with a 1-hour logging interval, from which the mean, maximum (max) and minimum (min) daily temperatures were calculated. The Freezing index (FI), defined as the integral of the temperatures below freezing during a given freezing season (approximately mid-October to April)——(Tuhkanen, 1980) was calculated for each station. Similarly, the Thaw Index,  $(TI)_{\tau}$  was calculated for positive temperatures (Harris, 1981). To characterize (Fig. 4) the thermal regime of the cave, the temperatures were divided into different groups and the respective percentages (%) per month were calculated (Fig. 4). These groups include positive temperatures, and temperatures ranging from 

Figure 4: <u>42-Twelve</u> years of cave and Armeña <u>Weather\_weather\_Station</u> <u>station (AWS)</u>-temperature <u>data</u> and characterization of the thermal state based on five temperature clusters. The dashed red line marks 0 °C; the black line represents the temperature trend. Note that the <u>short green lines</u> correspond to reconstructed datasets.

# 4. Results

## a. The old ice body

The ice body in the A294 cave <u>primarily</u> originates <u>mainly</u> from snow that <u>has-had</u> fallen or slid into the cave (Belmonte-Ribas et al., 2014; Sancho et al., 2018; Leunda et al., 2019). This deposit is characterized by the presence of cross-stratified ice and detrital layers formed by clasts and terrestrial plant macro-remains (Sancho et al.,

2018). Notably, there are no indications of congelation ice in this succession. The deposit studied in 2011 and 2015 (Sancho et al., 2018; Leunda et al., 2019) comprised five units separated by ice unconformities (Figs. 5a, b, c). These unconformities were created by sedimentary contacts parallel to the ice beds (paraconformities) and erosional contacts that cut across underlying beds (disconformities). Detrital layers correspond to decadal-scale ablation periods (Sancho et al., 2018). However, three of these unconformities represent longer periods with low snow and ice accumulation between ~5515\_-\_4945, ~4250\_-\_3810, and ~3155\_-\_2450 years cal BPago, indicating rather extended periods when the ice accumulation ceased, potentially due to drier and relatively warmer winters (Sancho et al., 2018). The ice accumulation ended about 1890 years ago-cal BP (Sancho et al., 2018).

Figure 5. a) Panoramic picture of the cliff of the A294 ice body showing its internal structure as identified by Sancho et al. (2018). b) and c) details of the ice stratigraphy including unconformities recognized in 2011 (Sancho et al., 2018).

In 2022, for the first time in 13 years of observation, the snow on the ramp disappeared completely at the beginning of fall allowing to sample directly the ice underneath the annual snow layer (Fig. S4). At the base of the ramp, the ice layers follow the ramp's slope and become more horizontal towards the top of the deposit. Two ablation pits (Fig. 6 and Fig. S4) formed by drips during the exceptionally warm summer of 2022 (Serrano-Notivoli et al., 2023) allowed the recovery of plant macro remains. The first sample from the middle part of the ramp provided a <sup>14</sup>C age ranging from 1509 to 1310 <u>yr\_cal\_-yr\_BP</u> (A294 R-mid, mean age 1409±99 (614±99 Common Era, CE)). The second sample, retrieved from about 1.5 m below the top of the deposit (approximately 15 m above the main ice body), ranges between 906 and 736 <u>yr\_cal\_yr\_BP</u> (A294 R-top; mean 821±85, (1202±85 CE)) (Fig. 6 and Table S1).

| Lab. code    | Sample<br>name | Material                  | Radiocarbon<br>age ( <sup>14</sup> C year<br>BP) | Radiocarbon<br>age 2 <sub>o</sub> ( <del>cal.</del> -yr<br><u>cal</u> BP) |
|--------------|----------------|---------------------------|--------------------------------------------------|---------------------------------------------------------------------------|
| D-AMS 049893 | A294 R-top     | Terrestrial plant remains | 908±24                                           | 736-906                                                                   |
| D-AMS 049894 | A294 R-mid     | Terrestrial plant remains | 1510±30                                          | 1310-1509                                                                 |

Table 1: Radiocarbon dates of terrestrial plant macrofossils of the A294 ice cave sampled in December 2022.

Figure 6. Ablation pit located close to the top of the ramp, horizontal detrital-rich ice layers, <u>and</u> location of sample A294 R-Top and its age. Photo taken on 27<sup>th</sup> December 2022. Note that the fresh snow dates from the fall, while in October 2022 the ramp was snow-free (<u>see also</u> Fig. S3).

Abundant blocks cover the ice body at the base of the ramp, forming a protalus rampart (Belmonte-Ribas et al., 2014), reaching a thickness of 50-60 cm (Fig. 1d). This accumulation implies that the upper section of the ice body has remained relatively stable for an extended period. The earliest cave survey from 1978 indicated indicates that the ice body was in contact with the entire cave perimeter, except for a small ablation pit hole and a narrow rimaye (Fig. S2). The current shape of the ice body in the lower part is characterized by an overhanging part, while the ice surface displays scallops ranging from decimeters to meters in scale (Fig. 2).

## 4.2 Climate and cave environment variability over the last 12 years

Analysis of the temperature data from the Armeña Weather weather Station-station (AWS)-depicts notable trends over the past 12 years. The mean winter temperature (WT) (Fig. 7a) shows a warming trend, with an increase of 0.07 °C a<sup>-1</sup> on average, resulting in a cumulative rise of 0.86 °C over the study period. Concurrently, the

number of freezing days during winter exhibits a slight decrease at a rate of 303 approximately 1 day a<sup>-1</sup> (Fig. 7b). The Freezing Index (FI) and the Thaw Index (TI) (Fig. 304 305 7c) reveal an increase and a marginal decrease, respectively. 306 In the A294 cave, sensors CH1 and CH2 are positioned at the top (around -23 m) and 307 bottom (approximately -33 m) of the ice body, respectively, while CH3 is located near the cave entrance (at roughly -6 m). Over the 12-year period, the mean temperature 308 displays negative values at CH1 (-0.45 °C) and CH2 (-0.64 °C), whereas CH3 recorded 309 a positive mean value (+1.91 °C). The mean annual cave air temperature increased 310 significantly at all sensors over the 12 years (Fig. 2): CH1 increased by 0.08 °C a<sup>-1</sup>, 311 totalling 1.07 °C; CH2 increased by 0.10 °C a<sup>-1</sup>, totalling 1.28 °C; CH3 increased by 312 313 0.13 °C a<sup>-1</sup>, summing up to 1.56 °C. The number of freezing days (Fig. 2b) decreased at CH1 (3.3 days a<sup>-1</sup>, 40 days in total), CH3 (1.3 days a<sup>-1</sup>, 16 days in total), an in CH2 314 (0.44 days a<sup>-1</sup>, 5.3 days in total). The record reveals a decreasing number of freezing 315 days for all calculated temperature ranges (Fig. 4). 316 317 The data from the AWS highlights a slight increase (decrease) in the FI and TI, respectively (Fig. 2c). Inside the cave, the CH2 and CH3 sensors exhibit the lowest 318 mean FI values (-370 and -339 °C hours per year, respectively), whereas CH1 show a 319 slightly higher mean value (-333 °C hours per year), which maybe is related to the 320 position of the sensor close to the ice deposit. In contrast, CH2 and CH3 are more 321 exposed to cold temperatures given their position. The TI shows slightly higher 322 values at CH1 compared to CH2 (158 vs. 122 °C hours per year), while CH3 presents 323 the highest values (962 °C hours per year), evidently influenced by the external 324

temperature.

Figure 7: a) Variation of winter mean temperature at Armeña Weather weather Station (AWS), and mean cave temperatures for each sensor during the open phase. b) Variation of the number of freezing days outside and inside of A294. c) Freezing and Thaw Lindexes (°C hours per year) at the AWS and in A294 during the study monitoring period.

In terms of precipitation (Fig. 8), the Armeña cirque received considerable monthly rainfall of more than 200 mm in October 2012, June 2015, November 2016, May 2017, November 2019, April and October 2020, and September 2021. The rainiest periods, spanning from April to November (mainly liquid precipitation), occurred between 2013 and 2016 (Fig. 8a). 2015 was particularly remarkable with 989 mm of rainfall. Similarly high values occurred between 2019 and 2021, with a remarkable 1312 mm of cumulated annual rainfall in 2020. These periods are also correlated with a higher frequency of rainy days. 2014 recorded the highest number of rain events (126), of which 46 and 51 events occurred during summer and fall, respectively. Years 2013 and 2020 were also notable, with 122 (46 in summer; 39 in fall) and 123 (39 in spring; 39 in summer; 44 in fall) rainfall events, respectively.

Extreme rainfall events (exceeding 80 mm) occurred at the end of October and beginning ofearly November in-2011 (102 and 91 mm), in November 2016 (83 and 86 mm), and in July and October 2017 (131 and 111 mm). The amount of rainfall and the number of rainy days increased during the monitoring period (Fig. 8a). The cumulative snow accumulation (Góriz meteorological station) was close to or exceeded 2 m in the winters of 2008-2009, 2009-2010, 2012-2013, 2014-2015, 2016-2017, and 2017-2018. In contrast, the winter of 2011-2012 stands-was characteriszed byout with the lowest snow accumulation, reaching a maximum of 80 cm.

Figure 8: a) Monthly rainfall in mm (light blue vertical bars) and the number of rainy days (dark blue line) and total rainfall (purple line) from April- to November (grey dashed lines) every year. b) Mean cave air temperature and Thaw Index (TI) during the closed phase measured atby sensors CH1 and CH2 compared to the number of rainy days (dark blue line) and total rainfall in mm (purple line).

Figure 8: a) Monthly rainfall in mm (April-November, light blue vertical bars) and the number of rainy days (dark blue line) and total rainfall (purple line) from every year. b) Mean cave air temperature and Thaw Index (TI) during the closed phase at sensors CH1 and CH2 compared to the number of rainy days (dark blue line) and total rainfall (purple line).

# 4.3 Cave dynamics and external climate connections

The cave sensors recorded pronounced seasonal fluctuations during the winter. CH3 and CH2 sensors exhibit the highest mean correlations (r) (n=12, 12 years) between the outside winter mean air temperature and the cave air temperature during the open phase, with values of 0.78 (p<0.001) and 0.72 (p<0.001), respectively. CH1 shows a slightly lower correlation of 0.69 (p<0.001). The highest correlation was observed during the coldest winters, such as in 2017-2018 (CH1: 0.80, p<0.001) and 2012-2013 (CH2: 0.86, p<0.001). CH3 often shows high correlations, for example in the years 2019-2020 and in the winters between 2011 and 2013 (0.88, p<0.001; 0.86, p<0.001; and 0.83, p<0.001), indicating a stronger influence of the outside temperature in those periods. Conversely, during the warmest winter (2016-2017), the cave sensors recorded the lowest correlations (CH1: 0.46, p<0.001; CH2: 0.46, p<0.001; CH3: 0.50, p<0.001). The influence of winter temperature during the open phase causes a thermal impact in the subsequent closed phase, particularly at the deepest sensor, CH2 (Fig. S54).

Rainfall exerts a considerable impact on the cave air temperature, leading to short temperature rises (around +1.2 °C) during the closed phase (Fig. 8b). The trends observed in cave temperature and TI at CH1 and CH2 during the closed phase show a strong similarity with the total rainfall recorded at the AWS (Fig. 8b). The absence of distinct pattern at CH3 (not shown) may relate to its location close to the upper cave entrance. The Thaw Index, the temperature during the closed phases and the total rainfall (with the exception of the 2014 and 2017 periods, for which no data is available) exhibit strong correlations, but are not statistically significant (rainfall vs. TI-CH1: 0.71, p=0.03; rainfall vs. TI-CH2: 0.83, p=0.006; rainfall vs. T-CH1: 0.65, p=0.05;

rainfall vs. TI-CH2: 0.82, p=0.0082). When removing the closed phase (due to the exceptionally cold winter of 2013), all correlations increase, and one (TI-CH2) becomes statistically significant (rainfall vs. TI-CH1: 0.76, p=0.03; rainfall vs. TI-CH2: 0.90, p=0.002; rainfall vs. T-CH1: 0.69, p=0.06; rainfall vs. T-CH2: 0.88, p=0.007). The data indicates that extreme rainfall events (exceeding 80 mm) trigger a more substantial temperature increase compared to what is typically observed during the closed phase. This is particularly evident in the data from October and November 2011 (Fig. S<sub>65</sub>).

#### 4.4 Ice Loss

Throughout the 12-year monitoring period, substantial changes in the ice body have become apparent (Fig. 9). These observations are supported by photographic evidence, field measurements, and comparisons with cave surveys dating back to the late 1970s. The most significant changes are related to the lateral retreat of the ice body from the northern (points 1 and 2) and the eastern (point 3) cave walls (Fig. 3).

A comparison of cave surveys (horizontal retreat) conducted on the North wall (point 1) from 1978 to 2012 indicates a mean ice retreat of ~480 cm since 1978 (ranging between approximately 220 cm to 800 cm, n=9), equalling an annual loss of about 14.3 cm. However, the most significant change was observed during the 2019 survey when sections of the ice had completely vanished between the North and South walls (Figs. 3c and 10\_-a, b, c, d, e, f). Between 2012 and 2019, the mean ice loss was approximately 1370 cm (ranging between ~1240 cm to 2100 cm, n=8), translating to an average loss of approximately 196 cm a<sup>-1</sup>. The Rock Ceorner (point 2) remained almost stable between 2008 and 2011 (Fig. 10a, b). Measurements taken by a laser distance meter between 2011 and 2023 (Fig. 10 c, d, e) revealed an ice loss of ~435 cm (retreat rate of ~62 cm a<sup>-1</sup>, n=1, between 2011and 2018) and ~610 cm (retreat rate of ~43 cm a<sup>-1</sup>, n=1, between 2018 and 2022, Fig. 10e, f). In 2023, the total ice loss was ~753 cm (rate ~143 cm a<sup>-1</sup>, n=3, between 2022 and 2023).

Concerning the East wall (point 3), the ice body was in contact with the wall until 2014 (Figs. 9a, b, c). Observations of seasonal ice stalagmites indicate that IS1 formed on the boulders at the cave floor from 2014 (Fig. 9c, July 2014), underlining the retreat of part of the ice deposit from the eastern East wall. Nevertheless, IS2 remained over the ice deposit (Fig. 9c).

Figure 9: Panorama photos of the A294 ice body between 2009 and 2022 (a, b, c, d, e, f). The ice body was in contact with the East wall (3) between 1978, such as the cave survey <u>indicates shows</u>, and 2014, when the ice body retreated. The retreat of ice body from the North wall (1) and the Rock <u>C</u>eorner (2) is also documented by photos showing the width of the snow ramp. Seasonal ice stalagmites (IS1 and 2).

Between 2014 and 2015, the ice retreated approximately 410 cm (n=2) due to the collapse of the overhanging sector (Figs. 9c, d). In 2018, IS2 formed on the boulders, as well as in 2022 (see Figs. 9e, f). The cave survey in 2019 indicates a mean ice loss of about 960 cm (retreat rate of ~166 cm a<sup>-1</sup>-\_n=3, 2015-2019). This rate is similar to the one calculated for the North wall (196 cm a<sup>-1</sup>; 2012-2019). In 2023, the ice front was 1300 cm (n=3) from the East wall (~100 cm a<sup>-1</sup>, 2019-2023). Lastly, vertical changes in the ice deposit based on stratigraphic comparisons (Sancho et al., 2018;

Leunda et al., 2019); Fig. S3) reveal an ice retreat of about 60 cm between 2011 and 2015, which corresponds to approximately ~15 cm a<sup>-1</sup>. The evolution of the overhanging shape was difficult to assess due to the continuous change in the cave floor morphology caused by rockfalls from the upper part of the deposit due as a result of ice retreat. A continuous decrease was observed from 2009 until 2015, when the overhanging ice collapsed. The presence of a warm airflow associated with a deeper karst system (e.g., Bertozzi et al., 2019), which would explain the overhanging shape, is ruled out as the CH2 sensor shows the coldest temperatures. Probably, this shape is related to ice sublimation associated with local air circulation, as suggested by the presence of scallops.

Figure 10: Changes and evolution of point 2 (Rock Ceorner) between 2008 and 2018. Red arrows show fix points.

## 5. Discussion

426

432433

436

438

445

447

#### 5. 1 Climate control on the melting rate

During winter, there is a strong correlation between cave air and outside air temperature, indicating the ingress of cold, dense air during open phases, leading to cave refrigeration. This open ventilation pattern is commonly observed in ice caves (e.g., Luetscher et al., 2008; Perşoiu et al., 2021; Wind et al., 2022). The correlation diminishes during warmer winters, such as in 2016-2017, implying reduced cave refrigeration. Additionally, the temperature recorded in A294 during open phases influences also the thermal regime during subsequent closed phases, similar to observations in other sag-type cave (Wind et al., 2022). Moreover, the positive temperatures recorded during closed phases indicate heat transfer from dripping points to the cave atmosphere consistent with the rainfall record, as evidenced by temperature increases following extreme rainfall events. Notably, during the exceptionally cold winter of 2012-2013, the cave was well refrigerated, resulting in low temperatures during the subsequent closed phase, while the impact of rainfall on cave temperature was almost negligible (Fig. 3b). This reduced influence of rainfall during the closed phase when the cave is well-refrigerated suggests that the heat transported by seepage water is exchanged primarily in-with the host rock and does not reach the cave atmosphere. Similarly, during snow-rich winters, the cave temperature during the closed phase may be impacted by the thermal inertia of snow accumulation in the cave. The temperature increase observed during the open phases in the cave (between ~1.07 and 1.56 °C) exceeded the increase of in winter temperature (~0.86 °C). Inside the cave, 2009-2010 marked the coldest winter of the monitoring period. In contrast, the winter of 2009-2010 was less cold and only 0.2 °C below the reference period 1971-2000, unlike 2011-2012 (third coldest winter of the 21st century) and 2017-2018 (seventh coldest winter since the beginning of 21st century in Spain) (AEMET, 2010, 2012, 2018). The analysis of the AWS climate data reveals that there were 41 days with temperatures below <u>-</u>4 °C during December, January, and February (D, J, F; the coldest months of the open phase). This is the highest number of days with temperatures below -4  $^{\circ}$ C in the entire external temperature record. The predominance of cold days during the winter of 2009–2010 likely contributed to the lowest mean temperature in the cave.

The calculated ice retreat is similar for the North wall (point 1) and the East wall (point 3). However, the retreat is lower for the Rock Corner Ccorner (point 2) and along the vertical axis. These differences are due to the fact that the Rock Corner Ccorner and the basal part of the ice ramp are supplied each year by snow (and previously by the formation of seasonal congelation ice). As a result, snow and ice must melt first every year before the old ice layers start to thaw. Our observations suggest that the continuous retreat of the ice body over the last decade is related to i) the steady increase in winter temperature, which has reduced the heat exchange (i.e. cooling) during the open phase, and ii) the increase in rainfall (amount and number of days) between April and November. The freezing capacity of the snow and water entering the cave seems to have diminished, promoting continuous melting.

### 5.2 Unprecedented ice melt since 6100 years ago

Following the Holocene Thermal Maximum (HTM, Fig. 11), during which the global mean surface temperature was 0.7 °C warmer than the median of the 19<sup>th</sup> century (Kaufman et al., 2020), the onset of the Neoglacial period (~6000-5000 years ago) was accompanied by large glacier advances in the Pyrenees (García-Ruiz et al., 2020). Those advances were associated with cooling periods in the North Atlantic region leading to a progressive temperature drop (Wanner et al., 2011; Bohleber, 2019). In the nearby Basa de la Mora lake (BSM, 3.6 km NW of the cave), the beginning of the Neoglacial is characterized by a drop in July temperatures of ~1.5 °C based on chironomids (Tarrats et al., 2018) (Fig. 11a).

# Código de campo cambiado

Figure 11: a) July temperatures reconstructed from chironomids obtained from the Basa de la Mora lake in the study area (Tarrats et al., 2018). b) Winter-spring temperatures reconstructed from chrysophyte cysts in Lake Redón (Pla and Catalan, 2005). c) High snow accumulation in A294: purple from Sancho et al. (2018), pink from this study. d) Number of Dryas octopetala macrofossils in the A294 ice body (Leunda et al., 2019). e) Advances of Monte Perdido and Troumouse glaciers (Gellatly et al., 1992; García-Ruiz et al., 2014, 2020). f) Mean annual temperature (red) and winter anomaly (blue) <u>5 years running average</u> from Pic du Midi de Bigorre (period 1882-2008) weather station (2860 m a.s.l., Bücher and Dessens, 1991; Dessens and Bücher, 1995). g) Glacier retreat in the Central Pyrenees (Marti et al., 2015). h) Annual precipitation (bluegrey), summer (red), and fall (orange) from a Torla weather station located 39 km NW of A294 at 1053 m a.s.l. i) Evolution of freezing days calculated from the Pic du Midi de Bigorre weather station. j) Ice loss (cm) in A294: lines show vertical loss (pink), Rock Corner Ceorner (blue, point 2), East wall (purple, point 3), and North wall (black, point 1). k) Changes in Pyrenean glacier area (ha) (Rico et al., 2017; Vidaller et al., 2021). I) Global cave ice loss - cumulative mass balance (m³) of ice caves (Kern and Perşoiu, 2013).

This temperature drop, and the relative stable winter-spring temperatures reconstructed from chrysophyte cysts at lake Redón (Pla and Catalan, 2005) during most of the Neoglacial (Fig. 11b), the cessation of tufa formed during part of the HTM (from 8700±134 to 6637±78 year cal- BP) in the San Bizién creek in the Cotiella massif (Belmonte, 2014), are coherent with the beginning of snow and ice accumulation in the A294 approximately 6100 years ago (Sancho et al., 2018) (Fig. 11c). In response to the relatively cold conditions following the HTM, an increase in the number of *Dryas octopetala* leaves found (Fig. 11d) within the ice body in A294 indicates the establishment of alpine meadows (dominated by *D. octopetala*), and a subsequent decline of the tree line ecotone (Leunda et al., 2019). Glaciers also responded to this cooling by advancing in the nearby cirques of Monte Perdido (García-Ruiz et al., 2020; Fig. 1c) (6900±800 <sup>36</sup>Cl yr BP) and Troumuse (Gellatly et al.,

480

483

488

1992) (6190\_-\_5735, 5905\_-\_5485 yrears cal- BP) (28 km to the Northwest) (Fig. 11e). During the Neoglacial, phases of high ice accumulation occurred at 6100 - 5515, 4945 -\_4250, 3810\_-\_3155, and 2450\_-\_=1890 vr\_cal- BP in A294 (Fig. 11c), while other advances occurred in the Monte Perdido glacier at ~3500±400, ~2500±300, and ~1100±100 36Cl yr cal BP, and during the Little Ice Age (LIA) (García-Ruiz et al., 2014). Sancho et al. (2018) and Leunda et al. (2019) suggested that the ice accumulation in the A294 ice cave ended in the Roman Period (RP), a well-known warm period oin the Iberian Peninsula and in the Central Pyrenees (Morellón et al., 2009; Martín-Puertas et al., 2010; Cisneros et al., 2016; Margaritelli et al., 2020; Bartolomé et al., 2024). Our new radiocarbon dates, however, indicate that the ice accumulation continued during the Dark Ages (DA) (1409±99 vr\_cal- BP), a cold period in the Pyrenees, (Bartolomé et al., 2024), while the final ice accumulation took place 821±85 years yr cal BP ago corresponding to the end of the Medieval Climate Anomaly (MCA). During the MCA, summer temperature reconstructions based on Pyrenean tree-rings-ring <u>data</u> show similar warm temperatures as those in the 20<sup>th</sup> Century (Büntgen et al., 2017), while the Monte Perdido glacier suffered from important melting episodes (Moreno et al., 2021). The age (821±85 yrears cal. BP) of the current top of the deposit, located ~15 m above the main ice body, suggests the cave could have been completely filled with snow and ice during the LIA. The lack of ice from the Little Ice Age (LIA), however, suggests two potential scenarios (Sancho et al., 2018): firstly, that ice may have formed during the LIA and then subsequently melted away. Alternatively, it is plausible that the cave entrance was blocked due to the intensified and recurrent snowfall characteristic of the LIA, thereby preventing snow from entering the cave. The current thick accumulation of debris on top the ice body at the foot of the ramp may have resulted from ice melt since the end of the LIA. This is likely given the increasing temperatures that affected the ice body near the entrance, while the morphology of the ice ramp has been preserved by seasonal snow accumulation at

495

499

501

506

511

513514

the cave entrance. We suggest that the top of the ice body has been affected by a

negative mass balance since the LIA, while the oldest layers of the deposit have only been affected more recently, as the oldest cave survey shows.

In the A294 ice cave, data about the past ice ice accumulation rates mass balance can be obtained from the analysis of the ice stratigraphy. Steep and vertical surfaces similar to those resulting from the current melting are absent in the ice succession. If a similar melting phase had occurred in the past, the ice stratigraphy would show truncated strata and high-angle unconformities due to accommodation following aggradation phases. The ice stratigraphy and the local climate reconstructions (Sancho et al., 2018; Tarrats et al., 2018; Leunda et al., 2019; García-Ruiz et al., 2020) therefore suggest that the ice melt in the last 6100 years was never as significant as it is today. This interpretation is consistent with the ongoing retreat of the Monte Perdido glacier and others Pyrenean glaciers (e.g., López-Moreno et al., 2016, 2019; Vidaller et al., 2021, 2023), which suggests that the current warming in the Pyrenees is the fastest and the most intense of the last 2000 years (Moreno et al., 2021), while current global temperatures are the highest in the last 6500 years (Kaufman et al., 2020).

### 5.3 The beginning of the end

The onset of the current ice retreat in A294 is difficult to determine. The 1978 cave survey indicates that the ice body covered almost the entire cave perimeter, except for a small pit hole and rimaye. The temperature (annual and winter) anomaly record of the Pyrenees (Midi de Bigorre station, at 2862 m a.s.l., Fig. 11f) shows a continuous temperature increase since 1880 (beginning of the record) with warm periods between 1890 and 1900, during the 1940's, around 1955-1970, and in recent decades (dashes rectangles, Fig. 11f). The Central Pyrenean glaciers (Marti et al., 2015 and references therein) show three main retreat phases since the end of the LIA (~1850 \_\_\_\_1900; ~1924 \_\_\_\_1965; 1980 - today, Fig. 11g), with slightly heterogeneous responses, which could be attributed to the generally larger influence of the local topography on smaller glaciers (Vidaller et al., 2021).

On the other hand, the annual regional precipitation anomaly series (1910-2013, 2535 m a.s.l.) for the Central Pyrenees shows a high inter-annual variability and a slightly negative and not significant trend of -0.6 °C decade<sup>-1</sup> (Pérez-Zanón et al., 2017). Unfortunately, the high-altitude meteorological stations in the nearby areas only started monitoring in 1950. Therefore, to evaluate rainy periods, we have used the nearest long-term precipitation series (Fig. 11h) located 39 km NW of the A294 at 1053 m a.s.l. (Torla station). A relatively warm period associated with an increase in summer rainfall led to a glacier retreat in the 1960s and an increase in fall precipitation until the year 1970 (dashed rectangle, Fig. 11h). This precipitation increase in fall was also registered at a regional scale (Pérez-Zanón et al., 2017). These observations, together with the 1978 cave survey, suggest that the observed ice retreat could have started as early as the 1960s.

> Between 1978 and 2012, the ice body in the A294 ice cave retreated, particularly at the North wall. After 1980 an important drop in freezing days is observed (Fig. 11i), as a consequence of a general increase in winter temperature. This situation led to a long-term snow decline below 2000 m a.s.l. (López-Moreno et al., 2020), which, together with an increase in rainfall during the warm season, favoured ice melt in the cave. The clear increase in melting rate between 2014 and 2019 (Fig. 11j) coincides with an important ice loss of up to 15 m (~6.1 m on average) on of the Monte Perdido glacier between 2011 and 2020 (Vidaller et al., 2021). The continuous loss of ice in A294 was also coincided with an important phase of glacier retreat in the Pyrenees (Fig. 11k) (Rico et al., 2017; Vidaller et al., 2021). -On a global scale, many ice caves are experiencing an important reduction in their ice volume (Fig. 11I) in response to the current climate warming (e.g., Luetscher et al., 2005; Kern and Persoiu, 2013; Serrano et al., 2018; Colucci and Guglielmin, 2019; Persoiu et al., 2021, Obleitner et al., 2024). However, the ice loss in A294 contrasts with other Pyrenean areas located in discontinuous mountain permafrost where the cave ice bodies are still protected from the current warming (Bartolomé et al., 2023).

> In terms of hydrological and environmental impactsRegarding the hydrological and ecological impact, the contribution to the local and regional hydrology of the from

the complete melting of all the ice in A294 and as well as that of all the subterranean ice masses in the Pyrenees to the local and regional hydrology, would be negligible, similar to the contribution of water if all Pyrenean glaciers melt, as calculated by that would be released were to melt(López-Moreno et al., (-(2020)). The ecological impact, it would primarily occur at the level of individual cave ecosystems. Ruiz-Blas et al. (2023) identified investigated the prokaryotic and eukaryotic microbial communities inhabiting within the ice in cave A294 across different depositional layers of ice deposition. By simulating temperature increases, the study mimicked in vivo climate change effects to assess microbial responses. This study simulated the in vivo effects of climate change by examining how a temperature increase would affect microbial populations. The results revealed molecular modifications associated with cellular adaptation under warmer conditions to higher temperatures.

-The A294 ice cave is an cleear example of the current <u>rapid</u> loss of paleoclimate information. Fortunately, some of this valuable information was recovered in time (Sancho et al., 2018; Leunda et al., 2019; Ruiz-Blas et al., 2023). Thus, we encourage the scientific community to study the valuable information stored in cave ice before <u>this these</u> paleoenvironmental archives <u>is are</u> lost forever.

### Conclusions

The investigation of A294 ice cave, based on environmental monitoring and ice melt tracking, as well as the stratigraphy and chronology of the deposit and its comparison with other <u>regional</u> paleoclimate records in the area, provides relevant information on the current ice melting and associated mechanisms.

Cave air temperature has risen by approximately 1.07 to 1.56 °C from 2009 to 2021. This warming is attributed to i) an increase in winter temperature, affecting the cave temperature during the open phase of refrigeration, and ii) an increase in rainfall

drippingseepage points water during the closed phase. 612 The increase in cave temperature and rainfall, along with reduced snowfall, has led 613 to a continuous retreat of the ice body. The <u>elevated\_rising\_</u>cave temperatures have 614 615 resulted in a diminished capacity to freeze water and snow entering the cave. In cave areas affected by seasonal snow and ice formation, retreat rates are lower (~15\_-43 616 cm  $a^{-1}$ ), whereas in other sectors the rates reached  $\simeq 196$  cm  $a^{-1}$ . 617 618 The new samples obtained in 2022 after the complete disappearance of the snow ramp indicate that ice accumulation was maintained during the Dark Ages and part 619 620 of the Medieval Climate Anomaly. The chronology, the ice stratigraphy, the ice 621 retreat under current climate conditions, the regional paleoclimate record in the area, as well as older cave surveys indicate that the current ice melt is the most 622 intense in the last 6100 years, when the deposit began to form. 623 624 At current and projected future rates of climate change, it is very likely that the current-melting will continue or even increase and that the ice deposit in A294 will 625 vanish within the next decade, although a portion of the ice may remain partially 626 preserved beneath the seasonal snow accumulation zone. 627 628 629 630 **Data availability** 631 Data are available from the authors upon request. 632 Supplement 633 634 **Author contributions** 635 CSa, ÁB, MB conceived the idea and designed the strategy. MB, ÁB, MLe performed 636 fieldwork, maintained the sensor sensors download and maintenance, and surveyed 637 a the cave survey in 2019. MB conducted data analyses and designed the figures. 638 JIL-M provided quantile temperature series and reviewed climatic data. CSa and AM 639

during late spring, summer, and early fall, which transports heat through via

obtained the funding. MB wrote the draft manuscript including all-inputs, suggestions and revisions from AM, ÁB, MLe, MLu, CSp, JIL-M, BO-U, JL-M.

# **Competing interests**

The contact author has declared that none of the authors has any competing interests.

# 645 646

652

654

# Acknowledgements

In memory of Carlos Sancho. This paper is a tribute to our dear friend and colleague, Dr. Carlos Sancho (Fig. 12), who passed away too soon. We extend our sincere gratitude to Jean Claude Gayet, Ramón Queraltó, Carles Pons (Asociación Cientifico-Espeleológica Cotiella, ACEC), Alexandra Bozonet, Reyes Giménez, David Serrano and Mario Bielsa for setting up the Armeña weather station and for their help during fieldwork. We thank the AEMET (Agencia Estatal de Meteorología) foe for providing the data from the Góriz station. We would like to thank the two anonymous reviewers for their thorough and constructive comments, which helped to improve the clarity and quality of thise manuscript.

Figure 12: Dr. Carlos Sancho during fieldwork in a cave.

# 656 657

### Financial support

This work was supported by the Spanish Government and the European Regional Development Funds (OPERA, SPYRIT, PYCACHU, ORCHESTRA CGL2009-10455/BTE, CTM2013-48639-C2-1-R, and CGL2016-77479-R, INTERREG-POCTEFA OPCC2 (ref.

EFA082/15), -and-ADAPYR (2019-2022) ref. EFA346/19 and P4CLIMA "Towards a

IPC-ES-LIFE PYRENEES4CLIMA, 101104957). 664 665 References 666 Agencia Estatal de Meteorología (AEMET): Resumen estacional climatológico. Invierno 2009-667 2010, Ministerio de Medio Ambiente y Medio Rural y Marino, Gobierno de España, 6 pp., 668 https://www.aemet.es/documentos/es/elclima/datos\_climat/resumenes\_climat/estaciona-669 les/2010/Est invierno 092010.pdf, 2010. 670 Agencia Estatal de Meteorología (AEMET) (2012): Resumen estacional climatológico. Invierno 2011–2012, Ministerio de Agricultura, Alimentación y Medio Ambiente, 7 pp., https://reposito-671 672 rio.aemet.es/bitstream/20.500.11765/1220/1/Est invierno 11 12.pdf, 2012. Agencia Estatal de Meteorología (AEMET) (2018): Resumen estacional climatológico. Invierno 673 2017–2018, Ministerio de Agricultura y Pesca, Alimentación y Medio Ambiente, 7 pp., 674 675 https://www.aemet.es/documentos/es/serviciosclimaticos/vigilancia clima/resumenes cli-676 mat/estacionales/2018/Est\_invierno\_17\_18.pdf, 2018. 677 Barabach, J. and Stasiewicz, A.: Ice Caves caves as Emerging emerging Research research 678 Objects-objects of the Climateclimate-Crisis-crisis Eraera, Permafrost and Periglacial Processes, 679 n/a, https://doi.org/10.1002/ppp.2288, n.d2025.-680 Bartolomé, M., Cazenave, G., Luetscher, M., Spötl, C., Gázquez, F., Belmonte, Á., Turchyn, A. V., 681 López-Moreno, J. I., and Moreno, A.: Mountain permafrost in the Central Pyrenees: insights 682 from the Devaux ice cave, The Cryosphere, 17, 477-497, https://doi.org/10.5194/tc-17-477-2023, 2023, 683 684 Bartolomé, M., Moreno, A., Sancho, C., Cacho, I., Stoll, H., Haghipour, N., Belmonte, Á., Spötl, 685 C., Hellstrom, J., Edwards, R. L., and Cheng, H.: Reconstructing hydroclimate changes over the 686 past 2500 years using speleothems from Pyrenean caves (NE Spain), Climate of the Past, 20, 467-494, https://doi.org/10.5194/cp-20-467-2024, 2024. 687 688 Beguería, S., Tomas-Burguera, M., Serrano-Notivoli, R., Peña-Angulo, D., Vicente-Serrano, S. 689 M., and González-Hidalgo, J.-C.: Gap Filling of Monthly Temperature Data and Its Effect on 690 Climatic Variability and Trends, Journal of Climate, 32, 7797–7821, https://doi.org/10.1175/JCLI-D-19-0244.1, 2019. 691 692 Belmonte, Ánchel: Geomorfología del macizo de Cotiella (Pirineo oscense): cartografía, evolución paleoambiental y dinámica actual, Universidad de Zaragoza, 581 pp., 2014. 693 694 Belmonte-Ribas, Á., Sancho, C., Moreno, A., Lopez-Martinez, J., and Bartolome, M.: Present-695 day environmental dynamics in ice cave a294, central pyrenees, spain, Geografia Fisica e Dinamica Quaternaria, 37, 131–140, https://doi.org/10.4461/GFDQ.2014.37.12, 2014. 696 697 Bertozzi, B., Pulvirenti, B., Colucci, R. R., and Di Sabatino, S.: On the interactions between 698 airflow and ice melting in ice caves: A novel methodology based on computational fluid 699 dynamics modeling, Science of The the Total Environment, 669, 322–332,

climate resilient cross-border mountain community in the Pyrenees" (ref. LIFE22-

663

https://doi.org/10.1016/j.scitotenv.2019.03.074, 2019.

- Biskaborn, B. K., Smith, S. L., Noetzli, J., Matthes, H., Vieira, G., Streletskiy, D. A., Schoeneich,
- P., Romanovsky, V. E., Lewkowicz, A. G., Abramov, A., Allard, M., Boike, J., Cable, W. L.,
- Christiansen, H. H., Delaloye, R., Diekmann, B., Drozdov, D., Etzelmüller, B., Grosse, G.,
- Guglielmin, M., Ingeman-Nielsen, T., Isaksen, K., Ishikawa, M., Johansson, M., Johannsson, H.,
- Joo, A., Kaverin, D., Kholodov, A., Konstantinov, P., Kröger, T., Lambiel, C., Lanckman, J.-P., Luo,
- D., Malkova, G., Meiklejohn, I., Moskalenko, N., Oliva, M., Phillips, M., Ramos, M., Sannel, A. B.
- 707 K., Sergeev, D., Seybold, C., Skryabin, P., Vasiliev, A., Wu, Q., Yoshikawa, K., Zheleznyak, M.,
- and Lantuit, H.: Permafrost is warming at a global scale, Nature Communications, 10, 264,
- https://doi.org/10.1038/s41467-018-08240-4, 2019.
- Blaauw, M.: Methods and code for 'classical' age-modelling of radiocarbon sequences,
- Quaternary Geochronology, 5, 512–518, https://doi.org/10.1016/j.quageo.2010.01.002, 2010.
- Bohleber, P.: Alpine Ice ice Cores cores as Climate climate and Environmental environmental
- Archivesarchives, in: Oxford Research Encyclopedia of Climate Science,
- https://doi.org/10.1093/acrefore/9780190228620.013.743, 2019.
- Bücher, A. and Dessens, J.: Secular Trend\_trend of Surface-surface Temperature temperature at
- an Elevated elevated Observatory observatory in the Pyrenees, Journal of Climate, 4, 859–868,
- https://doi.org/10.1175/1520-0442(1991)004<0859:STOSTA>2.0.CO;2, 1991.
- Büntgen, U., Krusic, P. J., Verstege, A., Sangüesa-Barreda, G., Wagner, S., Camarero, J. J.,
- Ljungqvist, F. C., Zorita, E., Oppenheimer, C., Konter, O., Tegel, W., Gärtner, H., Cherubini, P.,
- Reinig, F., and Esper, J.: New Treetree-Ring ring Evidence evidence from the Pyrenees rReveals
- Western Mediterranean Climate-climate Variability variability since Medieval Times times,
- Journal of Climate, 30, 5295–5318, https://doi.org/10.1175/JCLI-D-16-0526.1, 2017.
- Cisneros, M., Cacho, I., Frigola, J., Canals, M., Masqué, P., Martrat, B., Casado, M., Grimalt, J.
- O., Pena, L. D., Margaritelli, G., and Lirer, F.: Sea surface temperature variability in the central-
- western Mediterranean Sea during the last 2700 years: a multi-proxy and multi-record
- approach, Clim. Past, 12, 849–869, https://doi.org/10.5194/cp-12-849-2016, 2016.
- Colucci, R. R. and Guglielmin, M.: Climate change and rapid ice melt: Suggestions from abrupt
- permafrost degradation and ice melting in an alpine ice cave, Progress in Physical Geography:
- Earth and Environment, 0309133319846056, https://doi.org/10.1177/0309133319846056,
- 2019
- Dessens, J. and Bücher, A.: Changes in minimum and maximum temperatures at the Pic du
- Midi in relation with humidity and cloudiness, 1882–1984, Atmospheric Research, 37, 147–
- 162, https://doi.org/10.1016/0169-8095(94)00075-O, 1995.
- Feurdean, A., Perşoiu, A., Pazdur, A., and Onac, B. P.: Evaluating the palaeoecological potential
- of pollen recovered from ice in caves: A case study from Scărişoara Ice Cave, Romania, Review
- of Palaeobotany and Palynology, 165, 1–10, https://doi.org/10.1016/j.revpalbo.2011.01.007,
- 2011.
- García-Ruiz, J. M., Palacios, D., Andrés, N. de, Valero-Garcés, B. L., López-Moreno, J. I., and
- Sanjuán, Y.: Holocene and 'Little Ice Age' glacial activity in the Marboré Cirque, Monte Perdido
- Massif, Central Spanish Pyrenees, The Holocene, 24, 1439–1452,
- https://doi.org/10.1177/0959683614544053, 2014.

- García-Ruiz, J. M., Palacios, D., Andrés, N., and López-Moreno, J. I.: Neoglaciation in the
- Spanish Pyrenees: A multiproxy challenge, Journal of Mediterranean Geosciences, in press2,
- <u>21-36</u>, 2020.
- Gellatly, A. F., Grove, J. M., and Switsur, V. R.: Mid-Holocene glacial activity in the Pyrenees,
- The Holocene, 2, 266–270, https://doi.org/10.1177/095968369200200309, 1992.
- González Trueba, J. J., Moreno, R. M., Martínez de Pisón, E., and Serrano, E.: `Little Ice Age'
- glaciation and current glaciers in the Iberian Peninsula, The Holocene, 18, 551–568,
- https://doi.org/10.1177/0959683608089209, 2008.
- Hammer, O., Harper, D. A. T., and Ryan, P. D.: PAST: Paleontological statistics software package
- for education and data analysis. 4(1): 9pp., Palaeontologia Electronica, 4 (1), 9, 2001.
- Harris, S. A.: Climatic Relationships relationships of Permafrost Permafrost Zones in
- Areas areas of Low low Winter winter Snowsnow Covercover, Arctic, 34, 64–70, 1981.
- Heeb, B.: The Next next Generation generation of the DistoX Ceave Surveying surveying
- Instrumentinstrument. CREG J., 88, 5-8., 2014.
- IPCC: IPCC, 2021: Climate Change 2021: The Physical Science Basis. Contribution of Working
- Group I to the Sixth Assessment Report of the Intergovernmental Panel on Climate Change
- [Masson-Delmotte, V., P. Zhai, A. Pirani, S.L. Connors, C. Péan, S. Berger, N. Caud, Y. Chen, L.
- Goldfarb, M.I. Gomis, M. Huang, K. Leitzell, E. Lonnoy, J.B.R. Matthews, T.K. Maycock, T.
- Waterfield, O. Yelekçi, R. Yu, and B. Zhou (eds.)]. Cambridge University Press, Cambridge,
- United Kingdom and New York, NY, USA, 2391 pp., 2021.
- Kaufman, D., McKay, N., Routson, C., Erb, M., Dätwyler, C., Sommer, P. S., Heiri, O., and Davis,
- B.: Holocene global mean surface temperature, a multi-method reconstruction approach, Sci
- Data, 7, 201, https://doi.org/10.1038/s41597-020-0530-7, 2020.
- Kern, Z. and Perşoiu, A.: Cave ice the imminent loss of untapped mid-latitude cryospheric
- palaeoenvironmental archives, Quaternary Science Reviews, 67, 1–7,
- https://doi.org/10.1016/j.quascirev.2013.01.008, 2013.
- Leunda, M., González-Sampériz, P., Gil-Romera, G., Bartolomé, M., Belmonte-Ribas, Á., Gómez-
- García, D., Kaltenrieder, P., Rubiales, J. M., Schwörer, C., Tinner, W., Morales-Molino, C., and
- Sancho, C.: Ice cave reveals environmental forcing of long-term Pyrenean tree line dynamics,
- Journal of Ecology, 107, 814–828, https://doi.org/10.1111/1365-2745.13077, 2019.
- Lompar, M., Lalić, B., Dekić, L., and Petrić, M.: Filling Gaps gaps in Hourly hourly Air air
- Temperature temperature Data data Using using Debiased debiased ERA5 Datadata,
- Atmosphere, 10, 13, https://doi.org/10.3390/atmos10010013, 2019.
- T75 López-Moreno, J. I.: Recent <del>Variations variations of Snowpack snowpack Depth depth in the</del>
- Central Spanish Pyrenees, Arctic, Antarctic, and Alpine Research, 37, 253–260,
- https://doi.org/10.1657/1523-0430(2005)037[0253:RVOSDI]2.0.CO;2, 2005.
- T78 López-Moreno, J. I., Revuelto, J., Rico, I., Chueca-Cía, J., Julián, A., Serreta, A., Serrano, E.,
- Vicente-Serrano, S. M., Azorin-Molina, C., Alonso-González, E., and García-Ruiz, J. M.: Thinning
- of the Monte Perdido Glacier in the Spanish Pyrenees since 1981, The Cryosphere, 10, 681–
- 694, https://doi.org/10.5194/tc-10-681-2016, 2016.

- López-Moreno, J. I., Alonso-González, E., Monserrat, O., Del Río, L. M., Otero, J., Lapazaran, J.,
- Luzi, G., Dematteis, N., Serreta, A., Rico, I., Serrano-Cañadas, E., Bartolomé, M., Moreno, A.,
- Buisan, S., and Revuelto, J.: Ground-based remote-sensing techniques for diagnosis of the
- current state and recent evolution of the Monte Perdido Glacier, Spanish Pyrenees, J. Glaciol.,
- 65, 85–100, https://doi.org/10.1017/jog.2018.96, 2019.
- López-Moreno, J. I., García-Ruiz, J. M., Vicente-Serrano, S. M., Alonso-González, E., Revuelto-
- Benedí, J., Rico, I., Izagirre, E., and Beguería-Portugués, S.: Critical discussion of: "A farewell to
- glaciers: Ecosystem services loss in the Spanish Pyrenees," Journal of Environmental
- Management, 275, 111247, https://doi.org/10.1016/j.jenvman.2020.111247, 2020.
- Luetscher, M., Jeannin, P.-Y., and Haeberli, W.: Ice caves as an indicator of winter climate
- evolution: a case study from the Jura Mountains, The Holocene, 15, 982–993,
- https://doi.org/10.1191/0959683605hl872ra, 2005.
- Luetscher, M., Lismonde, B., and Jeannin, P.-Y.: Heat exchanges in the heterothermic zone of a
- karst system: Monlesi cave, Swiss Jura Mountains, Journal of Geophysical Research: Earth
- Surface, 113, https://doi.org/10.1029/2007JF000892, 2008.
- Margaritelli, G., Cacho, I., Català, A., Barra, M., Bellucci, L. G., Lubritto, C., Rettori, R., and Lirer,
- F.: Persistent warm Mediterranean surface waters during the Roman period, Sci Rep, 10,
- 10431, https://doi.org/10.1038/s41598-020-67281-2, 2020.
- Marti, R., Gascoin, S., Houet, T., Ribière, O., Laffly, D., Condom, T., Monnier, S., Schmutz, M.,
- Camerlynck, C., Tihay, J. P., Soubeyroux, J. M., and René, P.: Evolution of Ossoue Glacier
- (French Pyrenees) since the end of the Little Ice Age, The Cryosphere, 9, 1773–1795,
- https://doi.org/10.5194/tc-9-1773-2015, 2015.
- Martín-Puertas, C., Jiménez-Espejo, F., Martínez-Ruiz, F., Nieto-Moreno, V., Rodrigo, M., Mata,
- 805 M. P., and Valero-Garcés, B. L.: Late Holocene climate variability in the southwestern
- Mediterranean region: an integrated marine and terrestrial geochemical approach, Clim. Past,
- 6, 807–816, https://doi.org/10.5194/cp-6-807-2010, 2010.
- Morellón, M., Valero-Garcés, B., Vegas-Vilarrúbia, T., González-Sampériz, P., Romero, Ó.,
- Delgado-Huertas, A., Mata, P., Moreno, A., Rico, M., and Corella, J. P.: Lateglacial and Holocene
- palaeohydrology in the western Mediterranean region: The Lake Estanya record (NE Spain),
- Quaternary Science Reviews, 28, 2582–2599, 2009.
- Moreno, A., Bartolomé, M., López-Moreno, J. I., Pey, J., Corella, J. P., García-Orellana, J.,
- Sancho, C., Leunda, M., Gil-Romera, G., González-Sampériz, P., Pérez-Mejías, C., Navarro, F.,
- Otero-García, J., Lapazaran, J., Alonso-González, E., Cid, C., López-Martínez, J., Oliva-Urcia, B.,
- Faria, S. H., Sierra, M. J., Millán, R., Querol, X., Alastuey, A., and García-Ruíz, J. M.: The case of a
- southern European glacier which survived Roman and medieval warm periods but is
- disappearing under recent warming, The Cryosphere, 15, 1157–1172,
- https://doi.org/10.5194/tc-15-1157-2021, 2021.
- Obleitner, F., Trüssel, M., and Spötl, C.: Climate warming detected in caves of the European
- Alps, Sci Rep, 14, 27435, https://doi.org/10.1038/s41598-024-78658-y, 2024.
- Observatorio Pirenaico de Cambio Global: Observatorio Pirenaico de Cambio Global: Executive
- summary report OPCC2: Climate change in the Pyrenees: impacts, vulnerability and
- adaptation, 2018., 2018.

- Pérez-Zanón, N., Sigró, J., and Ashcroft, L.: Temperature and precipitation regional climate
- series over the central Pyrenees during 1910–2013, International Journal of Climatology, 37,
- 1922–1937, https://doi.org/10.1002/joc.4823, 2017.
- Perșoiu, A., Onac, B. P., Wynn, J. G., Blaauw, M., Ionita, M., and Hansson, M.: Holocene winter
- climate variability in Central and Eastern Europe, Scientific Reports, 7, 1196,
- https://doi.org/10.1038/s41598-017-01397-w, 2017.
- Perşoiu, A., Buzjak, N., Onaca, A., Pennos, C., Sotiriadis, Y., Ionita, M., Zachariadis, S., Styllas,
- 831 M., Kosutnik, J., Hegyi, A., and Butorac, V.: Record summer rains in 2019 led to massive loss of
- surface and cave ice in SE Europe, The Cryosphere, 15, 2383–2399, https://doi.org/10.5194/tc-
- 15-2383-2021, 2021.
- Pla, S. and Catalan, J.: Chrysophyte cysts from lake sediments reveal the submillennial
- winter/spring climate variability in the northwestern Mediterranean region throughout the
- Holocene, Climate Dynamics, 24, 263–278, https://doi.org/10.1007/s00382-004-0482-1, 2005.
- R Core Team: R: A Language and Environment for Statistical Computing. R Foundation for
- Statistical Computing, Vienna-, 2020.
- Racine, T. M. F., Reimer, P. J., and Spötl, C.: Multi-centennial mass balance of perennial ice
- deposits in Alpine caves mirrors the evolution of glaciers during the Late Holocene, Sci. Rep.,
- 12, 11374, https://doi.org/10.1038/s41598-022-15516-9, 2022a.
- Racine, T. M. F., Spötl, C., Reimer, P. J., and Čarga, J.: radiocarbon constraints on periods of
- positive cave ice mass balance during the last millennium, julian alps (nw slovenia)
- Radiocarbon constraints on periods of positive cave ice mass balance during the last
- millennium, Julian Alps (NW Slovenia), Radiocarbon, 1–24,
- https://doi.org/10.1017/RDC.2022.26, 2022b.
- Reimer, P. J., Austin, W. E. N., Bard, E., Bayliss, A., Blackwell, P. G., Ramsey, C. B., Butzin, M.,
- Cheng, H., Edwards, R. L., Friedrich, M., Grootes, P. M., Guilderson, T. P., Hajdas, I., Heaton, T.
- 849 J., Hogg, A. G., Hughen, K. A., Kromer, B., Manning, S. W., Muscheler, R., Palmer, J. G., Pearson,
- C., Plicht, J. van der, Reimer, R. W., Richards, D. A., Scott, E. M., Southon, J. R., Turney, C. S. M.,
- Wacker, L., Adolphi, F., Büntgen, U., Capano, M., Fahrni, S. M., Fogtmann-Schulz, A., Friedrich,
- R., Köhler, P., Kudsk, S., Miyake, F., Olsen, J., Reinig, F., Sakamoto, M., Sookdeo, A., and
- Talamo, S.: The IntCal20 Northern Hemisphere Radiocarbon radiocarbon Age age Calibration
- <u>calibration Curve curve</u> (0–55 cal k\_BP), Radiocarbon, 62, 725–757,
- https://doi.org/10.1017/RDC.2020.41, 2020.
- Rico, I., Izagirre, E., Serrano, E., and López-Moreno, J. I.: Superficie glaciar actual en los
- Pirineos: Una actualización para 2016, Pirineos, 172, 029,
- https://doi.org/10.3989/Pirineos.2017.172004, 2017.
- Ruiz-Blas, F., Muñoz-Hisado, V., Garcia-Lopez, E., Moreno, A., Bartolomé, M., Leunda, M.,
- Martinez-Alonso, E., Alcázar, A., and Cid, C.: The hidden microbial ecosystem in the perennial
- ice from a Pyrenean ice cave, Frontiers in Microbiology, 14, 2023.
- Sancho, C., Belmonte, Á., Bartolomé, M., Moreno, A., Leunda, M., and López-Martínez, J.:
- Middle-to-late Holocene palaeoenvironmental reconstruction from the A294 ice-cave record
- (Central Pyrenees, northern Spain), Earth and Planetary Science Letters, 484, 135–144,
- https://doi.org/10.1016/j.epsl.2017.12.027, 2018.

- Serrano, E., Gómez-Lende, M., Belmonte, Á., Sancho, C., Sánchez-Benítez, J., Bartolomé, M.,
- Leunda, M., Moreno, A., and Hivert, B.: Chapter 28—Ice Caves in Spain, in: Ice Caves, edited by:
- Perşoiu, A. and Lauritzen, S.-E., Elsevier, 625–655, https://doi.org/10.1016/B978-0-12-811739-
- 2.00028-0, 2018.
- Serrano-Notivoli, R., Tejedor, E., Sarricolea, P., Meseguer-Ruiz, O., de Luis, M., Saz, M. Á.,
- Longares, L. A., and Olcina, J.: Unprecedented warmth: A look at Spain's exceptional summer
- of 2022, Atmospheric Research, 293, 106931,
- https://doi.org/10.1016/j.atmosres.2023.106931, 2023.
- Stoffel, M., Luetscher, M., Bollschweiler, M., and Schlatter, F.: Evidence of NAO control on
- subsurface ice accumulation in a 1200 yr old cave-ice sequence, St. Livres ice cave, Switzerland,
- Quaternary Research, 72, 16–26, https://doi.org/10.1016/j.yqres.2009.03.002, 2009.
- Tarrats, P., Heiri, O., Valero-Garcés, B., Cañedo-Argüelles, M., Prat, N., Rieradevall, M., and
- González-Sampériz, P.: Chironomid-inferred Holocene temperature reconstruction in Basa de
- la Mora Lake (Central Pyrenees), The Holocene, 0959683618788662,
- https://doi.org/10.1177/0959683618788662, 2018.
- Tuhkanen, S.: Climatic parameters and indices in plant geography., Acta phytogeographica
- Suecica, 105, 1980.
- Vidaller, I., Revuelto, J., Izagirre, E., Rojas-Heredia, F., Alonso-González, E., Gascoin, S., René, P.,
- Berthier, E., Rico, I., Moreno, A., Serrano, E., Serreta, A., and López-Moreno, J. I.: Toward an
- <u>Iceice-Free-free Mountain-mountain Rangerange</u>: Demise of Pyrenean Glaciers glaciers During
- during 2011–2020, Geophysical Research Letters, 48, e2021GL094339,
- https://doi.org/10.1029/2021GL094339, 2021.
- Vidaller, I., Izagirre, E., del Rio, L. M., Alonso-González, E., Rojas-Heredia, F., Serrano, E.,
- Moreno, A., López-Moreno, J. I., and Revuelto, J.: The Aneto glacier's (Central Pyrenees)
- evolution from 1981 to 2022: ice loss observed from historic aerial image photogrammetry and
- remote sensing techniques, The Cryosphere, 17, 3177–3192, https://doi.org/10.5194/tc-17-
- 3177-2023, 2023.
- Wanner, H., Solomina, O., Grosjean, M., Ritz, S. P., and Jetel, M.: Structure and origin of
- Holocene cold events, Quaternary Science Reviews, 30, 3109–3123,
- https://doi.org/10.1016/j.quascirev.2011.07.010, 2011.
- Wind, M., Obleitner, F., Racine, T., and Spötl, C.: Multi-annual temperature evolution and
- implications for cave ice development in a sag-type ice cave in the Austrian Alps, The
- Cryosphere, 16, 3163–3179, https://doi.org/10.5194/tc-16-3163-2022, 2022.
- Žák, K., Richter, D. K., Filippi, M., Živor, R., Deininger, M., Mangini, A., and Scholz, D.: Coarsely
- crystalline cryogenic cave carbonate-– a new archive to estimate the Last Glacial
- minimum permafrost depth in Central Europe, Climate of the Past, 8, 1821–1837,
- https://doi.org/10.5194/cp-8-1821-2012, 2012.