# Peer review of "Unprecedent cave ice melt in the last 6100 years in"

_EGUsphere, 2025_

## Author Comment (AC1)

**Responses to Anonymous Referee #1**

*This study presents a detailed analysis of ice loss in A294 Ice Cave over the last 12 years (2009–2021), with implications for climate change impacts on cave ice deposits. The authors use a combination of temperature monitoring, precipitation data, and ice loss measurements from historical cave surveys, offering valuable insights into the unprecedented nature of recent melting. The authors successfully place their results within the broader framework of climate reconstructions and ice retreat trends in the Pyrenees.*

We thank referee #1 for reviewing the manuscript and comments. Below we respond to the suggestions (text in green).

*Suggestions for areas of improvements:*

*1) The temperature reconstruction method using quantile-based gap-filling should include a validation step against external datasets beyond Góriz and La Renclusa stations.*

The authors consider it unnecessary to test additional stations beyond those used in this study, given the very low percentage of missing data in the series (<1%) and the strong performance observed during the independent validation period. In fact, Barca et al. (2024) emphasized that the method used to fill temperature gaps has a minimal impact when data gaps are below 5%, even in analyses of extreme events.

*2) More details on uncertainties associated with ice retreat rates would be helpful. For example, specifying potential sources of error in survey comparisons (e.g., instrument precision, human factors) would strengthen confidence in the retreat estimates.*

In the case of the speleological survey from 2019, the precision associated with each measurement (n=68) is ±1.5 mm (Leica Disto X310). In contrast to clean congelation ice, where the laser can penetrate the ice over several meters (e.g., Bartolomé et al., 2023) our measurements indicate this is not the case in firn. The points from which the measurements were taken are well identified in the field, but potential errors may arise from the inclination of the laser line when taking the measurement. Normally, several measurements are taken to ensure that there are no major variations. This human error could result in inaccuracies of up to 5 cm, which, for distances greater than 10 m, would represent less than 0.5% error (measurements from Northern (1) and Ice Front (3), Fig. 3d in the manuscript), and slightly more for Rock Corner (2).

Luetscher et al. (2005) used a comparison of historical surveys to highlight changes over time, as it is a simple method to observe ice changes. Of course, surveys carried out by different individuals may vary slightly depending on the level of detail, the skill of the surveyor, and human measurement error. In fact, in the 1974 topography, we noticed an error in the orientation of the north arrow. However, the ice loss is so evident that even when considering only a rough outline of the cave, the measurements taken in successive surveys are consistent with those reported by Belmonte et al. and with the ice retreat measured in this study. We will add some sentences about the uncertainties of measurements.

*3) Some sections, particularly those discussing temperature trends and ice retreat, could benefit from a clearer distinction between observed data and modeled inferences. For instance, in section 5.1, the link between external temperature anomalies and ice melting rates could be more explicitly quantified. Including R-squared values or additional statistical measures would strengthen the argument.*

We agree on this point; however, due to the resolution of the measurements and the comparison of surveys (n=3), it was difficult to obtain the same year-by-year resolution for melt rates and the increase in temperature/precipitation. This limits our ability to perform correlations similar to those established with other parameters derived from the weather station and the cave's internal sensors. In this sense, the discussion on this topic could only be linked to what is addressed between lines 416 and 426 of the manuscript, in section 5.1, regarding melting rates depending on the sector of the cave. Without consistent resolution in both measurements and environmental parameters, it is challenging to establish any correlation beyond indicating that melting rates have increased in recent years. We will add a few lines in the Methods section specifying which annual rates were inferred from cave surveys and which were obtained from direct measurements inside the cave.

*4) Linking the cave's ice retreat to regional hydrology or ecosystem impacts could provide further relevance beyond cryosphere studies.*

Thank you for this suggestion. However, the retreat or even the complete disappearance of this or all ice caves in the Pyrenees, would not have a significant impact on the local and regional hydrology. In fact, López-Moreno et al., (2020) calculated the water storage capacity of Pyrenean glaciers (Rico et al., 2017) and estimated that their contribution represents only 0.02% of the 3000 hm³ storage capacity of the 13 major dams built in the Central Pyrenees (López-Moreno et al., 2008); 0.04% of the 1500 hm³ annual average of water stored as snow in the Central Spanish Pyrenees; and less than 0.01% of the 7300 hm³ long-term mean annual runoff (1950–2006) from the main Pyrenean tributaries (López-Moreno et al., 2011). Thus, the hydrological contribution of Pyrenean glaciers is minimal, and their disappearance would not impose a significant additional pressure on water resources. Ice caves represent only a very small percentage of the cryosphere in the Pyrenees.

Regarding the ecological impact, it primarily occurs at the cave ecosystem level. Ruiz-Blas et al. (2023) identified the prokaryotic and eukaryotic microbial communities inhabiting the ice in A294 at different ages of ice deposition. This study simulated the in vivo effects of climate change, examining how a 4°C temperature increase would affect microbial populations. The results revealed molecular modifications, identified several proteins and enzymes associated with cellular adaptation to higher temperatures, documented the influence of temperature rise on biogeochemical cycles, and detected proteins expressed at elevated temperatures that could serve as climate change indicators. All this information will be briefly included in Section 5.3 of the manuscript.

*Figures & visual presentation*

*Figure 3: The cave survey should have clearer legends to indicate differences in retreat rates across different sectors of the cave. In panel (d) the ice limit for 2011 is shown but not for 2012. Why there are no plan views for all ice deposit limits shown in (d)?*

We outline this suggestion below.

*"The cave survey should have clearer legends to indicate differences in retreat rates across different sectors of the cave."*

We will enlarge the legend and add more information. Additionally, in the topographic maps, we will enlarge the red lines that indicate the measured points.

*"In panel (d) the ice limit for 2011 is shown but not for 2012."*

The positions in Figure 3d have been approximately reconstructed based on the topographic surveys conducted in 1974, 2012, and 2019, as well as photographic sequences. To avoid overcrowding the figure with additional lines, and given that the position of the deposit in 2011 and 2012 was very similar, we will add a label "2011–2012" to the pink line.

*"Why there are no plan views for all ice deposit limits shown in (d)?"*

No additional topographic surveys were carried out during these monitoring years; only those from 1974, 2012, and the most recent one from 2019 exist. We conducted the 2019 survey after learning about the existence of the 1974 topography, considering it would be valuable to have a new survey for comparison purposes.

*Figure 4: I suggest using "Twelve years …" instead of "12 years…"*

We will change 12 years by twelve years

*Figure 8: (a) The caption is unclear. Light blue bars are visible in other months than April to November. The dark blue lines are intended to represent the number of rainy days (April to November), but the curve includes data from months outside this period. The same applies to the total rainfall. Why are the Y-axes titled "April-November" when the plots show wider intervals? Are the last two plots (nr of rainy days and total rainfall) depicted in Fig. 8a necessary, given that they are also shown in Fig. 8b and in panel (a) there is a lack of correlation?*

We will clarify the footnote. We will add dashed vertical lines in the figure to clearly separate the April–November periods, and we will double-check the dates for both series.

Regarding the second point, we plotted both curves together with the temperature during the closed phase and the Thaw Index, to visually show the increase in temperature (closed phase), and the rise in both the amount and frequency of precipitation during the monitored period. Given the inflow of liquid water into the cave during the closed phase is the main mechanism driving ice melt and the increase in cave temperature, we would like to keep the figure as it was.

*Figure 9: Enhance text visibility in panels (e) and (f) by employing larger white font.*

We will increase the font size of the text in figures (e) and (f).

*Figure 11 is highly informative, but it is quite dense. Can you think of some changes that might improve readability? For (h), orange (Fall) is not the best color as is almost impossible distinguish from red.*

We will adjust the colors for fall, summer, and annual in Fig. 11h to enhance readability. To simplify and reduce the number of curves, only the moving averages will be retained in Fig. 11f.

*Minor editorial issues:*

*Lines 80-82: Several additional factors influence cave temperature during the "closed" phase, not solely the heat exchange during winter. The sentence requires revision for clarity, considering the information provided in lines 82-85.*

We will rephrase the lines 80-82 taking into account the lines 82-85

*Line 183: The (dd/mm/aa) format is used, but at line 187, authors use the (dd/mm/yy) format. Choose and maintain one format consistently.*

We will check the consistency along the text and we will homogenize the format

*Line 594 (Acknowledgements): should be "… foe for .." or it should only be "for"*

We will delete "foe"

*References: There are minor formatting inconsistencies in the list, which should be checked for uniformity.*

We will correct the inconsistencies in the reference list.

**REFERENCES**

Barca, E., Guagliardi, I. & Caloiero, T. A methodological approach for filling the gap in extreme daily temperature data: an application in the Calabria region (Southern Italy), Theor. Appl. Climatol., 155, 7447–7461 (2024). https://doi.org/10.1007/s00704-024-05079-2

Bartolomé, M., Cazenave, G., Luetscher, M., Spötl, C., Gázquez, F., Belmonte, Á., Turchyn, A. V., López-Moreno, J. I., and Moreno, A.: Mountain permafrost in the Central Pyrenees: insights from the Devaux ice cave, The Cryosphere, 17, 477–497, https://doi.org/10.5194/tc-17-477-2023, 2023.

López-Moreno, J. I., Beniston, M., and García-Ruiz, J. M.: Environmental change and water management in the Pyrenees: Facts and future perspectives for Mediterranean mountains, Global and Planetary Change, 61, 300–312, https://doi.org/10.1016/j.gloplacha.2007.10.004, 2008.

López-Moreno, J. I., Vicente-Serrano, S. M., Moran-Tejeda, E., Zabalza, J., Lorenzo-Lacruz, J., and García-Ruiz, J. M.: Impact of climate evolution and land use changes on

water yield in the ebro basin, Hydrology and Earth System Sciences, 15, 311–322, https://doi.org/10.5194/hess-15-311-2011, 2011.

López-Moreno, J. I., García-Ruiz, J. M., Vicente-Serrano, S. M., Alonso-González, E., Revuelto-Benedí, J., Rico, I., Izagirre, E., and Beguería-Portugués, S.: Critical discussion of: "A farewell to glaciers: Ecosystem services loss in the Spanish Pyrenees," Journal of Environmental Management, 275, 111247, https://doi.org/10.1016/j.jenvman.2020.111247, 2020.

Luetscher, M., Jeannin, P.-Y., and Haeberli, W.: Ice caves as an indicator of winter climate evolution: a case study from the Jura Mountains, The Holocene, 15, 982–993, https://doi.org/10.1191/0959683605hl872ra, 2005.

Rico, I., Izagirre, E., Serrano, E., and López-Moreno, J. I.: Superficie glaciar actual en los Pirineos: Una actualización para 2016, Pirineos, 172, 029, https://doi.org/10.3989/Pirineos.2017.172004, 2017.

Ruiz-Blas, F., Muñoz-Hisado, V., Garcia-Lopez, E., Moreno, A., Bartolomé, M., Leunda, M., Martinez-Alonso, E., Alcázar, A., and Cid, C.: The hidden microbial ecosystem in the perennial ice from a Pyrenean ice cave, Frontiers in Microbiology, 14, 2023. https://doi.org/10.3389/fmicb.2023.1110091

---

## Author Comment (AC2)

**Responses to Anonymous Referee #2**

We thank referee #2 for the revisions. Below we respond to the comments (text in green).

In this paper, the authors indicate, based on several sets of data, observations and inferences, that cave ice melt in the central Pyrenees (Spain) is unprecedented over the past ~6000 years. The "message" of the paper can be broken down into two sections: 1) melting of ice during the past decades and 2) unprecedented melting during the Holocene. The authors use cave climate monitoring data to decipher the main factors responsible for ice melting (and/or accumulation) and use these in combination with mapping of the extent of ice during the past ~45 years to reconstruct the general retreat of ice. Further, the authors combine these observations with data from previous studies (aided by two additional [14]C ages) to show that the melting that occurred in modern times is unique in the history of the ice cave. I find the first part well supported by data, measurements and associated discussions, but cannot say the dame about the second part. Support for this thesis is given mainly by circumstantial observations, mostly at the end of section 5.2. As the manuscript stands now, the title and conclusions are not supported by the data and the discussions. I suggest the authors restrict their analysis to the modern (i.e., post 1978 melting) – there findings fit well with similar data from the Alps and the Balkans and give strong support to the usage of cave ice as indicators of melting cryosphere. The part dealing with the Holocene, however, is not well constrained and would require much better support to be considered for publications. I will restrict my comments to this section, as it is the one where most of the criticism would go.

So, first, reconstructing the extent of past ablation in a sedimentary sequence is difficult, as it implies usage of an equation with two unknowns: accumulation and ablation. It is impossible to derive a well-constrained (age-wise) melting period, based on observations of what is left behind.

➔ Obviously, reconstructing the mass balance is not possible, and we agree with the reviewer on the difficulty of reconstructing the melting periods from an ice sequence in a cave. However, we would like to clarify that at no point has this study attempted to reconstruct the ice mass balance in the strict sense, as defined by the equation:

$$\Delta m_{ice} = m_{new\_ice} - m_{melted\_ice}$$

where, mass variations ($\Delta m_{ice}$) in subsurface ice accumulations result from a difference between seasonal ice accumulation ($m_{new\_ice}$) and annual melting ($m_{melted\_ice}$). It is impossible to determine how much ice melted in the past (e.g., in m or m$^3$). However, there are sedimentary structures in this deposit that indicate the occurrence of ablation phases. Sancho et al. (2018) provide a detailed description of the deposit's stratigraphy — summarized in our manuscript (Results, 4.a) — including variations in accumulation rates inferred from radiocarbon dating, as well as the interpretation of the unconformities within the deposit. The term "mass balance" appears in two instances throughout the text. The first is in line 482, and although it is not possible to provide a precise value, the available data suggest — and it seems reasonable to infer — a clearly negative mass balance, as stated in the manuscript. The second instance refers to the stratigraphy of the deposit. To avoid potential confusion, we will remove the term in that context and replace it with "ice accumulation rate" in this second case.

The authors say that no periods of extended melting occurred in the past, based on the putative absence of debris layers, but these are clearly identifiable in the figures and also the same authors (Sancho et al., 2018) identified several such periods based on the presence of unconformities in the ice deposit (mentioned also in lines 224-228 in the current manuscript).

→ At no point in the manuscript do we state that extended melting did not occur in the past, nor do we claim that debris layers are absent. In fact, we explicitly mention the presence of debris layers (referred to as *detrital layers*). Furthermore, we identified (lines 219–233) three periods of reduced ice accumulation between the main unconformities recognized in the stratigraphy — which are interpreted as associated with ablation phases — and provide an interpretation of these features. In fact, we say that if a similar phase of retreat to the one currently observed had occurred in the past, the stratigraphy we see today in the deposit would be completely different, with truncated strata and high-angle unconformities formed by the accommodation of subsequent snow that would adapt to the geometry of the unconformities.

This is also indicated in the current manuscript (lines 155-156), the authors identifying changes in the internal structure of the ice sequence (which would contradict their later statements).

→ The response to this comment is discussed further below in this document.

While it is not clearly stated in the manuscript, it seems, based on the photos, that while the ice filled most of the cavity, the current retreat happens in a series of steps that combine lateral melting (retreat from the walls) followed by collapse of the overhanging flat surface, and again retreat.

→ We respectfully disagree with this comment. These aspects are addressed multiple times throughout the manuscript. First, the 1978 topographic survey indicates that ice filled the cave up to the elevations recorded at that time. In fact, Figures 3A and 3D (reconstruction of the cave's ice extent in plan view based on topographic data and field observations) show that the ice occupied the entire floor of the cave. This is explicitly stated in lines 253–255 of the manuscript. Second, regarding the current retreat, Section 4.4 discusses the recent melting and lateral retreat of the ice, including the collapse of an overhanging sector (lines 364–365), as well as the most recent retreat of the ice.

This retreat (10+) would have definitely destroyed possible layers indicating melting that would have formed in the past.

→ We do not agree with this comment. As can be observed, the lateral retreat is not altering the stratigraphy of the layers, since the direction of retreat is perpendicular to the exposed surface of the ice body. In fact, its preservation over time is evident when comparing the two age models (2011 and 2015). These age models support the extent of the main unconformities that affected the deposit in a similar way. Of course, there are minor discrepancies in the ice thickness between these unconformities and/or detrital layers, but the phases of higher and lower accumulation are consistent (cf. supplementary material). The thickness of ice between the unconformities is not constant, since the newly accumulated snow adapted to the morphology of the

previously ablated surface. Additionally, since snow entered the cave from the ramp down to the bottom, it is coherent to find slightly greater thicknesses in the central part of the deposit compared to the snow accumulated above the unconformities towards the ramp, where it could taper out.

Second, the new 14C ages seems to be derived from layered ice deposit located 15 m above the main one. It is not clear what relationship exists between the two, but if they belong to the same ice mass, 15 m of missing ice must have melted away sometimes in the past, thus far exceeding the current melting.

➔ We do not agree with the reviewer on this point. We have observed that the ice deposit continues above and is stratigraphically connected to the sequence previously described by Sancho et al. (2018). The new radiocarbon ages obtained are in correct stratigraphic order and consistent with the previous ones.
Below, we provide a photograph showing the continuity of the newly exposed ice deposit, revealed after the complete melting of the seasonal snow cover. These details are discussed in lines 235 and 245 of the manuscript. We can include this photo, or a selection of them, in the supplementary material if deemed necessary.

[Figure]

Another support for the unprecedented of the melting is given by the comparison of the two age models. It is not clear how this data supports the hypothesis (and why is tucked away in the supplementary material). The two sedimentary logs look quite different so it is difficult to understand what the authors wanted to say – perhaps that there is no vertical unconformity (lines 485-487) inside the ice mass? This is just absence of evidence.

➔ We do not agree with the reviewer's comment. At no point in the manuscript do we use the comparison of age models to claim that the current melting is unprecedented. The comparison of the age models is used to estimate the centimeters of ice lost due

to the vertical retreat of the deposit (lines 151–156), and to discuss the retreat rates observed at different locations within the cave (Section 5.1).

Nowhere do we state that a vertical unconformity exists within the ice deposit. In lines 483 to 485, we indicate that if a past retreat event similar to the current one had occurred, the present-day ice stratigraphy would show truncated strata and large unconformities — features that are absent in the observed stratigraphy of the deposit. Therefore, we firmly believe that the available field evidence supports our conclusion that the current cave ice melt is unprecedented in the last 6100 years.

---

## Author Response (AR2)

**Dear Editor,**

Thank you very much for forwarding the review of the new version of the manuscript. We appreciate the feedback and understand the reviewer's concern regarding the section of the unprecedented nature of the current ice melt.

Nonetheless, we maintain that the combination of stratigraphic evidence—characterized by the absence of large-scale angular unconformities and a consistent stratigraphy between the two separate visits—alongside with a consistent radiocarbon chronology and additional regional paleoclimate records, provides a robust basis to propose that the present ice retreat is likely the most significant since the formation of the deposit approximately 6100 cal yr BP. Although our data support the absence of an equivalent melting in the last 6100 yrs, this does not preclude anything to what may have happened in the early Holocene.

Furthermore, during our most recent field visit on September 6th, 2025, we observed that the ice deposit is now nearly gone (Fig 1). If a melting event of similar magnitude had occurred during the last 6100 years, it would have been impossible to sample the oldest ice located at the base, since those layers would have been substantially reduced or buried beneath blocks, as we observe today, preventing access to this ancient ice.

In addition, both local and global paleoclimate records indicate that current temperatures are the highest since, at least, the end of the Holocene Thermal Maximum, as we discuss in the manuscript (e.g. Pla and Catalan, 2005; Tarrats et al., 2018; Kaufman et al., 2020). For these reasons, we believe it would be inappropriate to remove what we consider one of the most scientifically relevant aspects of the manuscript—an interpretation that is well-supported by multiple lines of evidence. Without this section, the manuscript could be perceived as merely another case study of current ice loss without any implications for past climate conditions. The field evidence substantiates our conclusion that the current cave ice melt is unprecedented in the context of the past six millennia.

Figure 1: Top: sequence of photos (2018, 2022, and 2025) taken from the upper part of the ice deposit, showing the dramatic retreat of the ice (green line). The dashed red line marks the edge of the ice scarp. Bottom: sequence of photos (2011, 2014, and 2025) showing the deposit dramatic melt-down of the ice deposit in these years.

**References**

Kaufman, D., McKay, N., Routson, C., Erb, M., Dätwyler, C., Sommer, P. S., Heiri, O., and Davis, B.: Holocene global mean surface temperature, a multi-method reconstruction approach, Sci Data, 7, 201, https://doi.org/10.1038/s41597-020-0530-7, 2020.

Pla, S. and Catalan, J.: Chrysophyte cysts from lake sediments reveal the submillennial winter/spring climate variability in the northwestern Mediterranean region throughout the Holocene, Climate Dynamics, 24, 263–278, https://doi.org/10.1007/s00382-004-0482-1, 2005.

Tarrats, P., Heiri, O., Valero-Garcés, B., Cañedo-Argüelles, M., Prat, N., Rieradevall, M., and González-Sampériz, P.: Chironomid-inferred Holocene temperature reconstruction in Basa de la Mora Lake (Central Pyrenees), The Holocene, 0959683618788662, https://doi.org/10.1177/0959683618788662, 2018.